# Blood Lead Level as Marker of Increased Risk of Ovarian Cancer in BRCA1 Carriers

**DOI:** 10.3390/nu16091370

**Published:** 2024-04-30

**Authors:** Adam Kiljańczyk, Milena Matuszczak, Wojciech Marciniak, Róża Derkacz, Klaudia Stempa, Piotr Baszuk, Marta Bryśkiewicz, Krzysztof Lubiński, Cezary Cybulski, Tadeusz Dębniak, Jacek Gronwald, Tomasz Huzarski, Marcin R. Lener, Anna Jakubowska, Marek Szwiec, Małgorzata Stawicka-Niełacna, Dariusz Godlewski, Artur Prusaczyk, Andrzej Jasiewicz, Tomasz Kluz, Joanna Tomiczek-Szwiec, Ewa Kilar-Kobierzycka, Monika Siołek, Rafał Wiśniowski, Renata Posmyk, Joanna Jarkiewicz-Tretyn, Ping Sun, Rodney J. Scott, Steven A. Narod, Jan Lubiński

**Affiliations:** 1Department of Genetics and Pathology, International Hereditary Cancer Center, Pomeranian Medical University, ul. Unii Lubelskiej 1, 71-252 Szczecin, Poland; adam.kiljanczyk@pum.edu.pl (A.K.); milena.matuszczak@pum.edu.pl (M.M.); piotr.baszuk@pum.edu.pl (P.B.); tadeusz.debniak@pum.edu.pl (T.D.);; 2Read-Gene, Grzepnica, ul. Alabastrowa 8, 72-003 Dobra, Poland; 3Department of Clinical Genetics and Pathology, University of Zielona Góra, ul. Zyty 28, 65-046 Zielona Góra, Poland; gosiastawicka33@gmail.com; 4Department of Surgery and Oncology, University of Zielona Góra, Zyty 28, 65-046 Zielona Góra, Poland; 5OPEN, Kazimierza Wielkiego 24 Str., 61-863 Poznań, Poland; 6Medical and Diagnostic Center, Niklinowa 9, 08-110 Siedlce, Poland; 7Genetic Counseling Center, Subcarpatian Oncological Hospital, 18 Bielawskiego Str., 36-200 Brzozów, Poland; 8Department of Gynecology, Gynecology Oncology and Obstetrics, Institute of Medical Sciences, Medical College, Rzeszow University, Rejtana 16c, 35-959 Rzeszow, Poland; 9Department of Histology, Department of Biology and Genetics, Faculty of Medicine, University of Opole, 45-040 Opole, Poland; 10Department of Oncology, District Specialist Hospital, Leśna 27-29 Str., 58-100 Świdnica, Poland; 11Holycross Cancer Center, Artwińskiego 3 Str., 25-734 Kielce, Poland; 12Regional Oncology Hospital, Wyzwolenia 18 Str., 43-300 Bielsko Biała, Poland; 13Department of Clinical Genetics, Podlaskie Medical Center, 15-399 Bialystok, Poland; 14Non-Public Health Care Centre, Cancer Genetics Laboratory, 87-100 Toruń, Poland; 15Women’s College Research Institute, Women’s College Hospital, University of Toronto, Toronto, ON M5G 1N8, Canada; ping.sun@wchospital.ca (P.S.);; 16Medical Genetics, Hunter Medical Research Institute, Priority Research Centre for Cancer Research, Innovation and Translation, School of Biomedical Sciences and Pharmacy, Faculty of Health and Medicine, University of Newcastle, Pathology North, John Hunter Hospital, King and Auckland Streets, Newcastle, NSW 2300, Australia

**Keywords:** microelements, cancer, BRCA1, ovarian cancer

## Abstract

BRCA1 mutations substantially elevate the risks of breast and ovarian cancer. Various modifiers, including environmental factors, can influence cancer risk. Lead, a known carcinogen, has been associated with various cancers, but its impact on BRCA1 carriers remains unexplored. A cohort of 989 BRCA1 mutation carriers underwent genetic testing at the Pomeranian Medical University, Poland. Blood lead levels were measured using inductively coupled plasma mass spectrometry. Each subject was assigned to a category based on their tertile of blood lead. Cox regression analysis was used to assess cancer risk associations. Elevated blood lead levels (>13.6 μg/L) were associated with an increased risk of ovarian cancer (univariable: HR = 3.33; 95% CI: 1.23–9.00; *p* = 0.02; multivariable: HR = 2.10; 95% CI: 0.73–6.01; *p* = 0.17). No significant correlation was found with breast cancer risk. High blood lead levels are associated with increased risk of ovarian cancer in BRCA1 carriers, suggesting priority for preventive salpingo-oophorectomy. Potential risk reduction strategies include detoxification. Validation in diverse populations and exploration of detoxification methods for lowering lead levels are required.

## 1. Introduction

A BRCA1 mutation increases the lifetime risk of breast cancer with up to 70% and of ovarian cancer with up to 40% [1,2]. In addition to mutations, several modifiers predict the risk of cancer, including age, reproductive history (parity, age at first birth, and breastfeeding), surgical history, exogenous hormones (contraceptives and hormone replacement therapy), radiation exposure, and lifestyle factors (alcohol, smoking, and physical activity) [3]. Additional modifiers include dietary elements [4], including essential and nonessential elements. One of the nonessential elements is lead, classified by the IARC as a Group 2A carcinogen (probably carcinogenic to humans). Possible mechanisms whereby lead exposure may contribute to cancer include DNA damage by reactive oxygen species, disruption of DNA synthesis and repair, interference in cell cycle control, and alterations in the expression of cancer-related genes [5]. Some studies have associated lead levels with cancer of the lung, stomach, bladder, esophagus, brain, and kidneys [5,6,7,8,9,10]. In five studies, no significant correlation between lead levels and breast cancer or endometrial cancer was found [11,12,13,14,15]. There have been no studies examining the association of lead with the risk of cancer in BRCA1 mutation carriers. 

## 2. Materials and Methods

The study group included 989 adult women, who received genetic counselling and testing between 2011 and 2017 at the Clinical Hospitals of Pomeranian Medical University in Szczecin, Poland, or at an associated hospital or outpatient clinic.

At the first study visit, a fasting blood sample was taken from each subject for genetic testing for BRCA1 mutations. First, 10 mL of peripheral blood was collected for analysis from all study participants into a tube containing ethylenediaminetetraacetic acid (EDTA). Each blood sample was collected between 8 a.m. and 2 p.m. and stored at −80 °C until analysis. Subjects with a deleterious BRCA1 variant were included in the study.

These patients are usually offered genetic testing soon after diagnosis during an outpatient visit and are offered the opportunity to participate in other clinical studies. Medical charts were reviewed for date of diagnosis, date of birth (≤1965, 1965–1975, 1975–1985, >1985), age at enrollment (≤40/40–50/≥50), ovary removal surgery (yes/no), smoking status (never/current/former), contraceptive use (ever/never), diabetes (yes/no), dietary supplements (ever/never), hormonal replacement therapy (ever/never), and BMI (<18.5/18.5–24.9/25.0–29.9/≥30.0).

The study was conducted in accordance with the Declaration of Helsinki and with the approval of the Ethics Committee of the Pomeranian Medical University in Szczecin under number KB-0012/73/10 of 21 June 2010. All participants provided written informed consent.

### 2.1. Measurement of Blood Lead Levels

Blood samples were obtained from fasting subjects by venipuncture using a Vacutainer^®^ system, product number 368381 (Becton Dickinson, Plymouth, DEV, UK). Blood was carefully divided into new cryovials and subsequently frozen at −80 °C until analysis. The composition of the samples was analyzed utilizing inductively coupled plasma mass spectrometry (ICP-MS) employing a NexION 350D apparatus (PerkinElmer, Norfolk, VA, USA). The Kinetic Energy Discrimination (KED) mode was applied for elemental analysis, with rhodium as the internal standard to counteract instrument drift and matrix effects. Detailed specifications regarding the NexION 350D instrument parameters utilized in the measurements are available upon request. In the course of the analysis, the blood samples underwent a 40-fold dilution with a blank reagent (70 µL blood: 2730 µL buffer). The blank reagent employed comprised high-purity water (>18 MΩ), TMAH (AlfaAesar, Kandel, Germany), Triton X-100 (PerkinElmer, Shelton, CT, USA), EDTA (Merck, Darmstadt, Germany), and ethyl alcohol (Merck, Darmstadt, Germany).

The calibration curve standards were created through dilution of the 1000 µg/mL Lead Standard stock solution (PerkinElmer Pure Plus, Shelton, CT, USA) with the blank reagent. The calibration method employed was matrix-matched, consistently yielding correlation coefficients for the calibration curve exceeding 0.999. The precision and accuracy of the measurements underwent assessment utilizing certified reference materials (CRMs), including ClinChek^®^ Plasmonorm Whole Blood Level 1 (Recipe, Munich, Germany) and Seronorm Whole Blood Level 2 (Sero, Billingstad, Norway). Further technical specifics, plasma operational configurations, and mass spectrometer acquisition parameters are available upon request. Additionally, the testing laboratory actively engages in an independent external quality assessment program, QMEQAS (Quebec Multielement External Quality Assessment Scheme), administered by the Institut National de Santé Publique du Québec. 

### 2.2. Statistical Analysis

All study participants were assigned to one of three tertiles depending on their blood lead level. To estimate the hazard ratios (HRs) for cancer risk according to tertile, univariable and multivariable Cox proportional hazards regression analyses were performed. In multivariable models, the following variables were taken into consideration: lead level (tertile), year of birth, age at blood draw (<50 vs. ≥50), oral contraceptive use (yes/no), hormone replacement therapy use (yes/no), smoking history (yes/no), and BMI (<18.5/18.5–24.9/25.0–29.9/≥30.0).

The cumulative risks of breast and ovarian cancer were calculated from the age at blood draw to diagnosis of breast or ovarian cancer, death from another cause, or last follow-up. For estimating the risk of ovarian cancer, women with oophorectomy prior to blood draw were excluded, and patients with oophorectomy in the follow-up period were censored at the time of oophorectomy. For the analysis of breast cancer risk, oophorectomy was included as a time-dependent variable.

All statistical analyses were performed using SAS, version 9.4.

## 3. Results

The study group consisted of 989 women diagnosed with a BRCA1 mutation. All patients were cancer-free at the date of baseline. The patients were followed up from the date of blood test for an average of 7.52 years, during which time 173 new cancers at various sites occurred (121 cases of breast cancer, 29 cases of ovarian cancer, and 23 cancers at other sites). Due to missing data, we excluded 84 patients from the prospective analyses. The characteristics of the study group are shown in Table 1. The distributions of lead levels are presented in Figure 1.

### Ovarian Cancer

A total of 782 women were eligible for this analysis. Women with a blood lead level greater than 13.6 μg/L had a three-fold higher risk of ovarian cancer compared to those with a blood lead level in the reference tertile <9.6 μg/L (tertile 3 versus tertile 1: univariable: HR = 3.33; 95% CI: 1.23–9.00; *p* = 0.02; multivariable: HR = 2.10; 95% CI: 0.73–6.01; *p* = 0.17) (Table 2). The ten-year cumulative risk of ovarian cancer was 3.8% for those in tertile 1, 3.6% for those in tertile 2, and 7.9% for those in tertile 3. Figure 2 presents Kaplan-Meir curve for ovarian cancer risk for ten years after blood draw.

There was no significant correlation between lead level and breast cancer risk. Compared to those in the baseline tertile, the adjusted odds ratio for breast cancer was 1.12 (95% CI 0.70–1.79: *p* = 0.63) for those in the middle tertile and 1.04 (%CI: 0.63–1.71, *p* = 0.89) for those in the highest tertile.

There were 23 women who were diagnosed with another form of cancer, including bladder, cervix colon, kidney, leukemia, lung, salivary gland, urothelial, sarcoma, skin, thyroid, and pancreas. For these women, the hazard ratio for lead level and cancer in the highest versus lowest tertile was not significant.

## 4. Discussion

In this paper, we show that a high blood lead level is a marker for ovarian cancer in BRCA1 carriers. A blood lead level >13.6 μg/L (tertile 3) increased the risk of ovarian cancer risk by 3.3-fold (*p* = 0.02—univariable analysis) compared to a level in tertile 1. The multivariable results were not statistically significant but showed a tendency of increased ovarian cancer risk. The association between blood lead and breast cancer risk was not statistically significant. In our study, an increased risk of ovarian cancer was noted at relatively low blood levels compared to previous studies, which were based on occupational exposures to high doses of lead.

The different associations between lead levels and the risks of breast and ovarian cancers may be associated with the mechanism of carcinogenesis in BRCA1 carriers (Table 3).

It is very difficult to identify all the signaling pathways involved in the pathogenesis of breast or ovarian cancer, as numerous links between signaling pathways have been found. Considering date from the literature, lead affects eight major signaling pathways that are involved in the pathogenesis of breast or ovarian cancer, of which six signaling pathways are common. The known differences in the pathogenesis of breast and ovarian cancer involve five signaling pathways, of which two are affected by lead. The differences involve the following:-Lysophosphatidic acid (LPA) is involved in the carcinogenesis of ovarian cancer, but not breast cancer. LPA induces the proliferation, survival, drug resistance, invasion, opening of tight intercellular junctions and closing of gap junctions, cell migration, or metastasis of ovarian cancer cells. No direct effect of lead on this signaling pathway has been demonstrated [16].-Anti-Müllerian hormone (AMH) at physiological concentrations promotes the proliferation and survival of ovarian cancer cells; again, no direct effect of lead on this signaling pathway has been demonstrated [17,18].

To the best of our knowledge, lead has not been studied in series of ovarian cancers. However, large cohort studies have been performed on the association of blood lead levels and risk of breast cancer in North America, Sweden, and Italy [11]. Similarly to our results, the risk of breast cancer was not changed depending on lead levels. What is important is that in the above studies, low nonoccupational exposure was analyzed, and thus, the environmental conditions in the discussed studies and our cohort were similar, although nonidentical. Thus, it may be that the contribution of lead to breast carcinogenesis is lower and similar between BRCA1 carriers and noncarriers.

Our studies suggest that BRCA1 patients with elevated lead levels should not delay prophylactic oophorectomy, which has been correlated with better overall survival and distant disease-free survival [19]. In addition, all carriers with a blood level >13.6 ug/L (tertile 3) might consider the option of detoxification. To date, methods of detoxification from lead are known only for high levels of this metal. Methods for detoxifying from low levels of lead are unknown to this date, but it is reasonable to assume that such procedures might be established. The results of our study require validation in other ethnic groups and regions of the world, as well as with mutations in other predisposing genes.

## 5. Conclusions

High lead blood levels are associated with an increased risk of ovarian cancer risk in BRCA1 carriers. It is important that this association is confirmed in other studies. Our results put emphasis on developing a detoxification method for lowering levels of this element.

## 6. Patents

Based on the results presented in the article below, a patent application (Application ID P.446899) has been filed with the Polish Patent Office.

## Figures and Tables

**Figure 1 nutrients-16-01370-f001:**
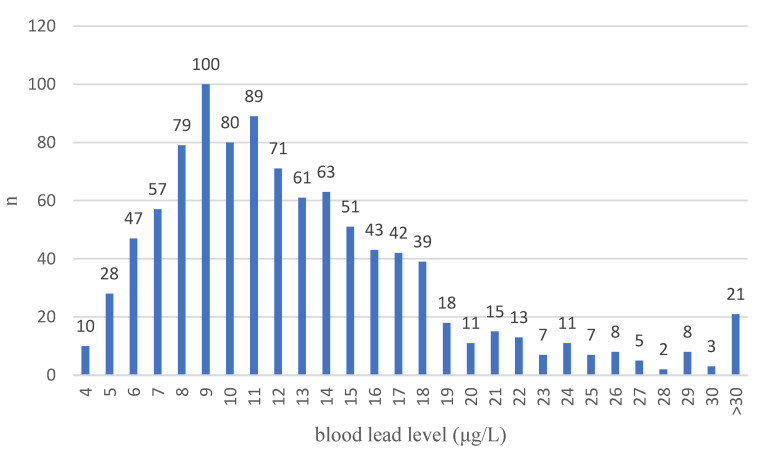
Distribution of lead level in cohort of BRCA1 carriers (*n* = 989).

**Figure 2 nutrients-16-01370-f002:**
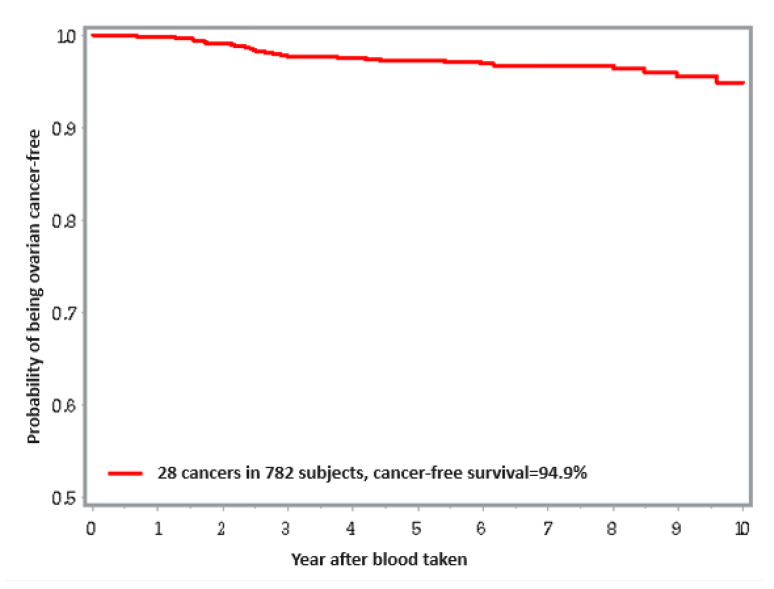
Kaplan–Meir curve for ovarian cancer risk for ten years after blood draw.

**Table 1 nutrients-16-01370-t001:** Group characteristics.

	*n* = 989
Age at enrollment	
<50	775 (78.36%)
≥50	214 (21.64%)
Smoking	
-never	720 (72.80%)
-ever	264 (26.70%)
-missing data	5 (0.50%)
Hormonal replacement therapy	
-never	720 (72.80%)
-ever	263 (26.59%)
-missing data	6 (0.61%)
Oophorectomy	
-no	413 (41.76%)
-yes	576 (58.24%)
-missing data	0 (0.00%)
Oral Contraceptive use	
-never	501 (50.66%)
-ever	481 (48.64%)
-missing data	7 (0.70%)
Diabetes	
-no	880 (88.98%)
-yes	62 (6.27%)
-missing data	47 (4.75%)
Body Mass Index	
<18.5	56 (5.66%)
18.5–24.9	553 (55.92%)
25.0–29.9	237 (23.96%)
≥30.0	95 (9.61%)
-missing data	48 (4.85%)
Dietary supplements usage	
-never	500 (50.56%)
-ever	489 (49.44%)
-missing data	0 (0.00%)
New cancer site (n = 174) (by the first cancer)	
breast	122 (70.11%)
ovarian	29 (16.67%)
bladder	2 (1.15%)
cervix	3 (1.72%)
colon	2 (1.15%)
kidney	1 (0.57%)
leukemia	2 (1.15%)
lung	3 (1.72%)
pancreas	1 (0.57%)
salivary gland	1 (0.57%)
sarcoma	1 (0.57%)
site unknown	1 (0.57%)
skin	1 (0.57%)
thyroid	3 (1.72%)
urothelial	1 (0.57%)
abdomen—CSU	1 (0.57%)

**Table 2 nutrients-16-01370-t002:** Hazard ratios for ovarian cancer according to lead level.

Variables	Ovarian Cases/ Total	Univariate HR (95% CI) P	Multivariate * HR (95% CI) P
Lead level			
≤9.6 μg/L	5/260	1	1
9.6–13.6 μg/L	6/261	1.12 (0.34–3.69) 0.85	0.98 (0.29–3.25) 0.97
>13.6 μg/L	18/261	3.33 (1.23–9.00) 0.02	2.10 (0.73–6.01) 0.17
Total	29/782		
Date of birth			
≤1965	10/101	1	1
1965–1975	9/164	0.49 (0.20–1.22) 0.13	1.43 (0.07–28.1) 0.82
1975–1985	9/328	0.25 (0.10–0.64) 0.003	0.44 (0.02–11.0) 0.62
>1985	1/189	0.06 (0.01–0.50) 0.006	0.09 (0.00–3.73) 0.22
Age at blood draw (years)			
≤40	14/556	1	1
40–50	5/129	1.53 (0.55–4.23) 0.42	0.44 (0.12–1.66) 0.22
>50	10/97	4.49 (1.99–10.1) 0.0003	1.27 (0.05–30.5) 0.88
Oral contraceptive use			
No	18/374	1	1
Yes	11/402	0.54 (0.25–1.14) 0.10	0.82 (0.35–1.91) 0.65
Missing	0/6		
Hormone replacement therapy			
No	26/662	1	1
Yes	3/154	0.40 (0.12–1.32) 0.13	0.34 (0.10–1.17) 0.09
Missing	0/6		
Smoking			
No	12/447	1	1
Current	7/176	1.46 (0.58–3.71) 0.42	1.27 (0.49–3.30) 0.63
Former	10/154	2.53 (1.09–5.85) 0.03	2.35 (0.99–5.59) 0.05
BMI at blood draw			
≤median (23.0)	11/396	1	1
>median (23.0)	16/339	1.70 (0.79–3.65) 0.18	0.98 (0.42–2.29) 0.96
Missing	2/47		

* Adjusted by all the variables listed in the left column. Women with oophorectomy prior to blood draw excluded. Censored at oophorectomy. Patients with no lead level measurement excluded.

**Table 3 nutrients-16-01370-t003:** Known carcinogenic mechanisms in breast and ovarian cancer with BRCA1 and lead interactions.

Mechanism	*BRCA1* Interactions	Breast Cancer	Ovarian Cancer	Lead Interactions
NF-kB (signaling path)	No	Yes	Yes	Yes
MAPK (signaling path)	Yes	Yes	Yes	Yes
ErbB (signaling path)	No	Yes	Yes	
AMH (signaling path)	No	No	Yes	
LPA (signaling path)	Yes	No	Yes	
PI3K (signaling path)	Yes	Yes	Yes	No
Estrogen Receptors (signaling path)	Yes	ERα+	ERβ+	
D1-CDK4/6-RB (signaling path)	Yes	Yes	No	Yes
FGF (signaling path)	Yes	Yes	No	
EGF (signaling path)	No	Yes	Yes	Yes
VEGF (signaling path)	Yes	No	Yes	Yes
SRC	No	Yes	Yes	
JAK	Yes	Yes	Yes	
HER2	No	Yes	Yes	
IGF-1 (signaling path)	Yes	Yes	Yes	No
NOTCH (signaling path)	Yes	Yes	Yes	Yes
E-cadherin-integrin	No	Yes	Yes	

## Data Availability

Data supporting the results presented are available from the authors upon request from any interested researchers.

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
