# Peer review of "Blood Lead Level as Marker of Increased Risk of Ovarian Cancer in BRCA1 Carriers"

_nutrients, 2024, doi:10.3390/nu16091370_

Round 1

Reviewer 1 Report

Comments and Suggestions for Authors

The study entitled "Blood lead level as a marker of increased risk of ovarian cancer in BRCA1 carriers" by Adam KiljaÅ„czyk et al. shows the association between blood lead levels and the risk of ovarian cancer in BRCA1 mutation carriers.

Conducted on 989 women, the research found that those with blood lead levels above 13.6 μg/L faced a three-fold increase in the risk of developing ovarian cancer compared to those in the lowest tertile of lead exposure.

Their results are crucial as they introduce a new variable in the management and prevention strategies for BRCA1 carriers.

Congratulations to the authors of the manuscript. I think it is very clear and well-written, I only suggest to incorporate this study (PMID: 35534308) that investigates the prognosis and oncological outcomes of BRCA-mutated patients treated with bilateral salpingo-oophorectomy, to increase the quality of your manuscipt and expand the discussion section which is a bit short.

Author Response

The study entitled "Blood lead level as a marker of increased risk of ovarian cancer in BRCA1 carriers" by Adam KiljaÅ„czyk et al. shows the association between blood lead levels and the risk of ovarian cancer in BRCA1 mutation carriers.

Conducted on 989 women, the research found that those with blood lead levels above 13.6 μg/L faced a three-fold increase in the risk of developing ovarian cancer compared to those in the lowest tertile of lead exposure.

Their results are crucial as they introduce a new variable in the management and prevention strategies for BRCA1 carriers.

Congratulations to the authors of the manuscript. I think it is very clear and well-written, I only suggest to incorporate this study (PMID: 35534308) that investigates the prognosis and oncological outcomes of BRCA-mutated patients treated with bilateral salpingo-oophorectomy, to increase the quality of your manuscipt and expand the discussion section which is a bit short.

-aforementioned paper added in discussion

-quality of figure 2 improved, changed events to cancers and last survival to cancer-free survival.

-specific number of patients in 21-30 ug/L lead level range added.

-corrected percentage mismatch and patient count in table 1

-Materials and Methods section paraphrased to avoid repetition (we used the same cohort in a series of papers)

-a section of Patents, Funding, Institutional Review Board Statement, Iinformed Consent Statement, Data Availability Statement and Conflicts of interest paraphrased.

-several typos fixed

Reviewer 2 Report

Comments and Suggestions for Authors

The communication article demonstrates a possibility that high blood lead level is associated with ovarian cancer.

Figures may be improved with higher resolution. Especially, Figure 2 is not clear enough what the 28 events mean in the figure legend.

In Figure 1, it seems to be interesting that the blood lead level (21-30 ug/L) has higher distribution. It would be better to have the span of the blood lead level as one as in the 4-20 ug/L.

Careful proofreading is needed. 

Author Response

The communication article demonstrates a possibility that high blood lead level is associated with ovarian cancer.

Figures may be improved with higher resolution. Especially, Figure 2 is not clear enough what the 28 events mean in the figure legend.

-quality of figure 2 improved, changed events to cancers and last survival to cancer-free survival.

In Figure 1, it seems to be interesting that the blood lead level (21-30 ug/L) has higher distribution. It would be better to have the span of the blood lead level as one as in the 4-20 ug/L.

-specific number of patients in 21-30 ug/L lead level range added.

Careful proofreading is needed.

-corrected percentage mismatch and patient count in table 1

-Materials and Methods section paraphrased to avoid repetition (we used the same cohort in a series of papers)

-a section of Patents, Funding, Institutional Review Board Statement, Iinformed Consent Statement, Data Availability Statement and Conflicts of interest paraphrased.

-several typos fixed

Reviewer 3 Report

Comments and Suggestions for Authors

It’s a veryequilibrated study,very clear exposed and also an original idea.The chapters are clear structured and the conclusions are simple and clear.The English language is fluent and The bibliography is appropriate cited.

Author Response

It’s a very equilibrated study, very clear exposed and also an original idea. The chapters are clear structured and the conclusions are simple and clear. The English language is fluent and The bibliography is appropriate cited.

-quality of figure 2 improved, changed events to cancers and last survival to cancer-free survival.

-specific number of patients in 21-30 ug/L lead level range added.

-corrected percentage mismatch and patient count in table 1

-Materials and Methods section paraphrased to avoid repetition (we used the same cohort in a series of papers)

-a section of Patents, Funding, Institutional Review Board Statement, Iinformed Consent Statement, Data Availability Statement and Conflicts of interest paraphrased.

-several typos fixed